# The potential of elastic/polarization lidars to retrieve extinction profiles

Elina Giannakaki[1,2], Panos Kokkalis[3,4], Eleni Marinou[4,5], Nikolaos S. Bartsotas[6], Vassilis Amiridis[4], Albert Ansmann[7], Mika Komppula[2]

[1]Department of Environmental Physics and Meteorology, University of Athens, Athens, 15784, Greece

[2]Finnish Meteorological Institute, Atmospheric Research Centre of Eastern Finland, Kuopio, 70211, Finland

[3]Physics Department, Faculty of Science, Kuwait University, Kuwait

[4]Institute for Astronomy, Astrophysics, Space Applications and Remote Sensing, National Observatory of Athens, Athens 15236, Greece

[5]Institute of Atmospheric Physics, German Aerospace Center (DLR), Oberpfaffenhofen, 82234, Germany

[6]Atmospheric Modeling and Weather Forecasting Group, Department of Physics, National and Kapodistrian University of Athens, Greece

[7]Leibniz Institute for Tropospheric Research (TROPOS), Leipzig, 04318, Germany

*Correspondence to*: Elina Giannakaki (elina@phys.uoa.gr)

**Abstract.** A new method, called ElEx, is proposed for the estimation of extinction coefficient lidar profiles using only the information provided by the elastic and polarization channels of a lidar system. The method is applicable both during day-time and night-time lidar measurements under well-defined aerosol mixtures. ElEx uses the particle backscatter profiles at 532 nm and the vertically resolved particle linear depolarization ratio measurements at the same wavelength. The particle linear depolarization ratio and the lidar ratio values of pure aerosol types are also taken from literature. The total extinction profile is then estimated and compared well with Raman retrievals. In this study, ElEx was applied in an aerosol mixture of marine and dust particles at Finokalia station during CHARADMEx campaign. Any difference between ElEx and Raman extinction profiles indicates that the non-dust component could be probably attributed to polluted marine or polluted continental aerosols. Comparison with sun-photometric aerosol optical depth observations is performed as well during daytime. Differences in the total aerosol optical depth is varying between 1.2 and 72% and is attributed to the limited ability of the lidar to correctly represent the aerosol optical properties in the near range due to overlap problem.

## 1 Introduction

Aerosols play an important role in the atmospheric radiation budget (IPCC, 2013). Depending on the aerosol type, they can absorb or scatter the incoming and outgoing radiation, warming or cooling the atmosphere and depending on their size and

composition, they can act as condensation nuclei, modifying cloud physical and radiative properties (Kauffman et al., 2002). However, climate forcing by tropospheric aerosols remain one of the largest uncertainties in climate variability and climate change studies. Pure types of aerosols can be categorized roughly as mineral dust, sea salt, volcanic, carbonaceous, or sulfate aerosols originating from various natural and anthropogenic sources.

Several lidar studies have revealed that a broad variety of aerosol mixtures occur in the European continent (e.g. Balis et al., 2004, Papayannis et al. 2005). The mixing occurs because of the relatively long pathways of air masses across different aerosol source regions before the detection over the European continent. Sea-salt particles are near spherical and non-absorbing, whereas dust particles are non-spherical and show considerable absorption. Thus mixing of either marine aerosol or absorbing aerosol or both with dust particles may result to different optical properties. The measured optical properties is then resulted

from the contribution of each aerosol type to the total aerosol load.

Lidar measurements provide vertical profiling of various particle properties with high spatial and temporal resolution, and based on the optical properties the identification of different aerosol types becomes feasible. The determination of the extinction-to-backscatter ratio (the so-called lidar ratio) profile is possible using the Raman-lidar technique (Ansmann, 1992). The lidar ratio contains information on the aerosol type, since it depends on the index of refraction and on the size of particles.

Many studies have demonstrated that the lidar ratio is a quantity valuable for aerosol characterization (Ansmann et al., 2002; Müller et al., 2007; Mattis et al., 2004; Amiridis et al., 2005; Giannakaki et al., 2010, Giannakaki et al., 2016; Groβ et al, 2011; Groβ et al, 2013; Groβ et al, 2015). However, the majority of aerosol lidars, including the recently launched Cloud-Aerosol Lidar and Pathfinder Satellite Observations (CALIPSO) lidar are so-called elastic-backscatter lidars. Such lidars allow us to retrieve only the particle backscatter coefficient (Klett, 1981). Elastic backscatter lidars are often equipped with polarization

measurements.

Recently, the polarization lidar technique has been additionally used to separate the desert dust aerosol component from other aerosol components (Tesche et al., 2009). Desert dust causes high depolarization of backscattered light, whereas typical non-desert aerosol mixtures, like marine aerosols, lead to very low depolarization. The technique is the base of POLIPHON algorithm  for ground –based lidars (Ansmann et al., 2019) and has also been applied to CALIPSO aerosol profiles either on

selected case studies (Giannakaki et al., 2011) or on a statistical basis (Marinou et al., 2017).

In this contribution, we propose a method to determine the extinction coefficient profile using only the elastic and polarization lidar channels at 532 nm. The method has been first suggested by Giannakaki et al. (2017) and further applied by Ansmann et al. (2017). In this contribution, we fully outline the methodology providing an extended sensitivity analysis along with the main advantages and limitations of it. The methodology is limited to cases where only two basic aerosol types are observed

and the mixing state of the atmosphere is well known. When the above mention criteria are met, extinction coefficient profiles can be retrieved with non-Raman lidars, even during daytime and with low time resolution windows of 1 hour or less. In case of more complicated aerosol mixture inside the planetary boundary layer, our method is still valid and applicable to free tropospheric aerosol layers.

## 2 Methodology

### 2.1 Elastic Extinction Retrieval: ElEx methodology

A new method, called ElEx [= Elastic Extinction], is proposed for the estimation of extinction coefficient lidar profiles using only the information provided by the elastic and polarization channels of a lidar system.

### 2.2.1 Backscatter coefficient and particle depolarization profile

At the first step, we retrieve the backscatter coefficient ($\beta_t$) at 532 nm. The lidar equation for the backscatter coefficient at the emitted wavelengths is solved following the Klett-Fernald retrieval methods (Klett, 1981). For the calibration of the profile of the measured 532 nm elastic backscatter signal, pure Rayleigh signals are simulated based on actual temperature and pressure profiles from numerical weather forecast data or actual nearby radiosonde observations. The measured 532 nm signals are then fitted to the Rayleigh signal profile in the aerosol-free middle to upper troposphere. The solution is possible only under the assumption of a constant with height relation between aerosol extinction and backscatter coefficient (lidar ratio), The assumed lidar ratio can be selected either from climatological studies or, if available, from night time measurements. Additional information provided from dust models, backward trajectories, sunphotometers and/or satellite retrievals may be helpful for the right selection of the initial lidar ratio value. ElEx also uses the particle depolarization ratio at 532 nm. The particle depolarization ratio is computed from the volume depolarization ratio by means of the determined particle backscatter coefficient (Freudenthaler et al., 2009).

### 2.2.2 Separation of the backscatter profile

At the second step we need to decompose our profile to its aerosol components, assuming that only two aerosol types are observed. The procedure to separate the aerosol mixture into two components starts from the equation of the particle depolarization ratio:

$$\delta_p = \frac{\beta_1^\perp + \beta_2^\perp}{\beta_1^\parallel + \beta_2^\perp} \qquad (1)$$

$\beta^\perp$ and $\beta^\parallel$ are so-called cross- and parallel-polarized particle backscatter coefficients that can in principle be computed from the lidar return signals detected with the cross-polarized and parallel-polarized signal channels.

As shown by Tesche et al. (2009), the separation of the two aerosol components is possible through the simple equation 2:

$$\beta_1 = \beta_t \frac{(\delta_t - \delta_2)(1 + \delta_1)}{(\delta_1 - \delta_2)(1 + \delta_t)} \qquad (2)$$

In order to perform the decomposition of the backscatter profile the two aerosol types should be distinguishable in terms of the particle depolarization ratio. This mean that the decomposition is possible using the information of the strongly depolarizing dust and a second aerosol type with less depolarizing ability, like marine, pollution or even biomass burning aerosols.

From the profile of the total particle backscatter coefficient $\beta_t$ and the backscatter coefficient of the 1st aerosol type $\beta_1$ we calculate the remaining backscatter coefficient of the 2nd aerosol type.

$$\beta_2 = \beta_t - \beta_1 \qquad (3)$$

### 2.2.3 Estimation of Elastic Extinction profile

The separation of the backscatter profile give us the opportunity to calculate the total extinction coefficient by applying the correct lidar ratio to each aerosol component.

$$a_t = a_1 + a_2 = \beta_1 \cdot lr_1 + \beta_2 \cdot lr_2 \quad (4)$$

This is only possible in case of the mixture of two aerosol types that are well known in terms of lidar ratio. ElEx is presented in the flowchart of Figure 1. Type 1 and 2 could be any types that can be distinguishable is terms of depolarization and lidar ratio values. Since ElEx methodology is strongly dependent on the selection of pure lidar ratio and particle depolarization values we briefly present the aerosol types that are mostly observed in Europe along with their intensive optical properties.

Continental polluted aerosols are typically small and do not significantly depolarize the backscattered light ($\delta_{aer}^{532} = 0.04 \pm 0.04$)
(Heese, et al., 2016), and due to the high carbon content, these particles reveal high lidar ratios (Giannakaki, et al., 2010). On the contrary, clean continental type differentiates from the polluted continental type due to its less light absorbing properties. The clean continental type shows low depolarizing ability with values lower than 0.07 (Omar, et al., 2009) and low lidar ratio values, i.e., 20–40 sr (Ansmann, et al., 2001; Giannakaki, et al., 2010).

In the absence of significant transport of continental aerosols, particles over the remote oceans are largely of marine origin
(Prospero, et. al., 1989).The sea-salt particles feature a predominant coarse mode, however, they are spherical in humid conditions and weakly absorbing, in contrast to the dust particles. Therefore, they yield low particle lidar ratio values, are almost non depolarizing and exhibit low particle depolarizing ratio values (Burton, et al., 2013; Dawson, et al., 2015). This aerosol type is mainly identifiable by the low particle lidar ratio, i.e., 15–25 sr at 532nm (Burton, et al., 2012).

Desert areas around the world emit huge quantities of dust aerosols which also actually extend considerably over adjacent
regions, such as oceans (Jaenicke, et al., 1978) and can be transported over very long distances (Prospero, et al., 1989; Papayannis, et al., 2008; Mona, et al., 2012). The optical properties of dust particles are considerably different from the other types, thus making them easy to identify, especially in the absence of aerosol mixtures. However, it needs to be taken into account that the dust optical properties depend on the source region and the transport pattern (Valenzuela, et al., 2014), which is a source of variability mainly detected in the lidar ratio (Nisantzi, et al., 2015). Recently, Mamouri et al. (2013) showed that
dust originating from the Arabian Desert produced significantly lower lidar ratio values (34–39 sr at 532nm) than respective values (50–60 sr at 532nm) from western Saharan dust particles.

Biomass burning is a major global source of atmospheric aerosols. Generally, smoke particles are relatively small, spherical, and highly absorbing that produce low depolarization and large lidar ratios (Amiridis, et al., 2009; Baars, et al., 2012; Nicolae, et al., 2013; Giannakaki, et al., 2016). The optical properties of smoke particles may vary due to the vegetation type of the
emitting source, the combustion type (smouldering or flaming fires), and atmospheric conditions (Balis, et al., 2003).

Furthermore, the particles are susceptible to changes of their optical properties during their lifetime in the atmosphere (Nicolae, et al., 2013). The similarities of the physical characteristics of smoke particles and continental particles result in similar optical properties, making these types difficult to distinguish.

Volcanoes are another important source of atmospheric aerosols. Volcanic eruptions eject great amounts of material in the atmosphere (tephra), while the fraction smaller than 2mm is labeled as volcanic ash. Most of these aerosols will settle only a few tens of kilometres away from the volcano but smaller particles can travel thousands of kilometres and affect wider areas (Mattis, et al., 2010; Sicard, et al., 2012; Papayannis, et al., 2012; Kokkalis, et al., 2013; Pappalardo, et al., 2013). The optical properties of volcanic ash aerosols is generally similar to the one of desert dust, as was shown by Ansmann et al. (2011) and Wiegner et al. (2012) for fresh ash with particle linear depolarization ratios reaching 0.37 and lidar ratios of 50–65 sr. Aged volcanic particles as observed by Papayannis et al. (2012) indicate higher sphericity less non-sphericity with depolarization ratio values of about 0.1–0.25 and lidar ratios for 355nm within the range 55–67 sr and for 532nm 76–89 sr.

ElEx is not limited to nighttime Raman observations, and thus is also applicable to daytime lidar measurements as long as the observed aerosol types have different particle depolarization ratios to permit for an accurate separation between them. The more the particle depolarization ratio between the two aerosol types differ the better the separation we achieve. The accuracy of the extinction coefficient is also depended on the knowledge of the correct lidar ratio. For example the intensive properties of marine and Saharan dust particles are already well defined, while smoke particles are characterized by large variability especially on lidar ratio values. In case of Raman extinction profiles availability the purity of the non-dust component can be additionally checked. In stations with more complicated aerosol mixtures within the planetary boundary layer the methodology is possible to be applied to free tropospheric aerosol layers. This is particularly useful in case of free tropospheric layers of dust or volcanic particles with smoke or pollution.

## 2.2 Description of the lidar system and lidar data processing

The main instrument of this paper is the Polly[XT] lidar as it is described by Althausen et al. (2009) and Engelmann et al. (2016). An overview of PollyNET can be found in Baars et al. (2016). Polly[XT] works with a Nd:YAG laser, emitting pulses at 1064, 532 and 355 nm, with a repetition frequency of 20 Hz. The receiver consists of a Newtonian telescope, with a diameter of 300 mm and a field of view of 1 mrad. Photomultiplier tubes are used for the detection of the elastically backscattered photons at 355, 532 and 1064 nm, as well as for the inelastically backscattered photons at 387 and 607 nm, which correspond to the Raman shift by nitrogen molecules at 355 and 532 nm, respectively. Additionally, the cross-polarized component of radiation at 355 and 532 nm is detected, allowing for the determination of the particle linear depolarization ratio (also called depolarization ratio). In this study, the polarization channels permit us to identify non-spherical dust particles from the near spherical marine aerosols. The Polly[XT] Arielle system used in this study has been provided by the TROPOS Institute.

The vertical resolution of the signal profiles is 30 m and the raw data are typically stored as 30 s average values. Data were collected on the web page of PollyNET (http://polly.tropos.de) where the "quicklooks" of all measurements are available.

Lidar data was collected during CHARADMExp campaign (Characterization of Aerosol mixtures of Dust And Marine origin) that took place from the 20th of June 2014 until the 20th of July 2014. CHARADMExp was an experimental campaign of ESA, implemented by the National Observatory of Athens (NOA), aimed to characterize dust and marine particles along with their mixtures (http://charadmexp.gr). The site for the campaign is the monitoring ACTRIS station of Finokalia in Greece as presented in Fig. 2. Finokalia station is located at a remote coastal site in the northeast of the island Crete, Greece, in the Eastern Mediterranean (35.338°N, 25.670°E, 252 asl). The station is located at a top hill, facing the sea within a sector from 270° to 90°. No touristic or other human activities can be found at a distance shorter than 15 km within the aforementioned sector. The site is affected by marine and dust particles by 95% of the time (Mihalopoulos et al., 1997; Kouvarakis et al., 2000).

For the purposes of this study backscatter coefficient profiles are calculated via Fernald–Klett method (Fernald, 1984; Klett, 1981). The method requires independent information on the lidar ratio and the reference value of the particle backscatter coefficient. Afterwards, the calibrated depolarization ratio profiles at 532 nm, are calculated (Freudenthaler et al. 2009). An overlap correction was applied on the basis of a simple technique proposed by Wandinger and Ansmann (2002). Thus, the aerosol profiles are retrieved down to 500 m. Extinction coefficient profiles at 532 nm are also retrieved based on the Raman method (Ansmann et al., 1992) and are only used for validation purposes of the proposed methodology at a later stage in this study.

## 2.3 Models

Four-day backward trajectories are calculated using the Hybrid Single-Particle Lagrangian Integrated Trajectory model (HYSPLIT) to gather information about the origin of the observed aerosols and the synoptic patterns corresponding to the measurements. The HYSPLIT 4 model is a complete system for computing simple trajectories to complex dispersion and deposition simulations using either puff or particle approaches. A discussion of the model is given by Draxler and Hess (1997) and Draxler and Hess (1998). The simulations were performed using the GDAS meteorical data. Backward trajectories were computed for several altitudes for the CHARADMExp campaign period, confirming that the origin of the air masses arriving over our site, were from Saharan region. The BSC-DREAM8b model is additionally used to verify the presence of Saharan dust, indicated from the trajectory analysis. The BSC-DREAM8b model described the desert dust emissions and transport (Nickovic et al., 2001; Pérez et al., 2006; Basart et al., 2012). Moreover, sea salt emissions and transport are described with the atmospheric model RAMS/ICLAMS (Solomos et al., 2011). The model is a further developed version of RAMS 6.0 (Pielke et al., 1992) that allows for a fully prognostic treatment of the sea salt particles and their life cycle in the atmosphere. The simulations were used to verify the presence of marine aerosols in the atmosphere during the campaign and specifically during the day under study.

## 3 Results

### 3.1 CHARADMExp Lidar measurements

Range corrected signal (RCS) at 1064 nm as a function of time and height is presented in Figure 3a. The aerosol load inside the Planetary Boundary Layer (PBL) was enhanced throughout the campaign period, while the free tropospheric aerosol load seems to vary with time. Volume depolarization ratio (Fig. 3b) highlight the presence of several depolarizing layers during the campaign, indicating the existence of non-spherical particles. Backward trajectory analysis have confirmed that the origin of these non – spherical particles originates from Saharan region. The lidar observations show that the dust events developed in

the Free Troposphere to end up within the PBL after some days. Thus, RCS within the PBL appears larger after each dust event. Specifically, there are three intense dust events: the first event took place during 17th and 18th of June 2014 at heights up to 5 km. Volume depolarization ratio was maximum during these days with values up to 35%. The second dust event started at noon of 24th of June 2014 in the height of 2 km and within a day the layer has reached up to 5.5 km. This dust event lasted more than 4 days, while several smaller (in terms of intensity) followed. The third dust event started on 5th of July 2014. This

dust layer appeared thin with strong volume depolarization ratio and remained thin but not stable in height for almost 3 days. Then this thin layer seemed that expanded up to 6 km.  The third event was followed by clouds. The remaining days of the campaign appeared with low volume depolarization ratio indicating almost spherical particles throughout the atmospheric column.

### 3.2 Application to a case study: Dust and marine aerosol mixture

We apply ElEx methodology in the measurement performed at the night of 28th of June 2014. The case study was selected based on several criteria: (1) the direction of the air masses indicate dust transport from Saharan region and possible mixture of marine aerosols as well, (2) values of the linear particle depolarization ratio is within 5 and 30 % reveal again possible mixture of dust and marine aerosols, (3) large aerosol load that leads to large SNR which make Raman retrievals possible and (4) the availability of cimel sun-photometric data during that day. Criteria (3) gives us the opportunity to compare our

methodology to concurrent results of extinction profile with the well-established Raman method, while criteria (4) is used to compare our day-time extinction profiles retrieval with an independent instrument.

Four days backward trajectories arriving at Finokalia station on 28th of June 2014 at 02:00 UTC are presented in Figure 4. The trajectories are computed for arrival heights of 500 (red), 2000 (blue) and 4000 m (green) to cover height range of the observed layers that we recognize in coherent structures of height time displays of the range-corrected lidar signal (Figure 3a). The

trajectory analysis reveals that is highly possible that the air masses carry marine aerosols at Finokalia station on the day under study, especially in lower heights (500 m arrival height). Trajectory analysis suggest also that dust particles are advected over Finokalia station. Mass concentration profiles performed with RAMS/ICLAMS and DREAM/BSC models are presented in Figure 5 and also confirm the presence of marine aerosol at heights up to 2.5 km and dust particles up to 5 km. Mean sea-salt

concentration reach 60 μg/m$^3$ at the first 200 m, decreasing with height. On the contrary the dust mean mass concentrations appear to increase with height, taking the maximum value of 70 μg/m$^3$ at ~1300m.

The application of ElEx methodology was performed in 3 hours averaged lidar signals (00:00-03:00 UTC). Backscatter coefficient, using the Klett-Fernald method with initial lidar ratio of 40 sr (Fernald, 1984; Klett, 1981) was retrieved, and is presented in Figure 6 (a) with the black line. A double layer structure was observed; the first layer occurred between 1.4 and 2.8 km and the second one, less intense in terms of aerosol load, between 3 and 5 km. The particle depolarization ratio was measured between 15% and 30%, throughout the atmospheric column, as shown in the same Figure with the green line. The separation of the two aerosol types was applied, and the resolved dust and marine backscatter coefficients are presented in Figure 6 (b). Dust backscatter coefficient, is following the double layer structure, while marine particles contribute less to the total aerosol load. Appropriate lidar ratios were then used, for the estimation of the separated extinction coefficient (see Table 1 and Figure 2), and the estimation of the final retrieval of total extinction coefficient. At the same time frame, Raman extinction coefficient profiles were computed. The results are presented in Figure 6 (c). Comparison between Raman and the proposed methodology shows a very good agreement. In addition, the derived extinction coefficient using the proposed methodology has less noise because is based on the elastic signal. The retrieval of extinction profiles with low temporal resolution is one of the advantage of ElEx methodology.

A small difference is observed between the two extinction coefficient profiles at heights greater than 2.8 km. We believe that the aerosol particles of anthropogenic origin are more probable at these heights, rather than marine aerosols. Anthropogenic aerosols, do not depolarize the light and thus the value of 0.05 that was used fits well for the separation of dust and non-dust particles in general. However, for the retrieval of extinction coefficient profile the aerosol type depended lidar ratio is crucial. Thus, for heights above 2.8 km, a value of 60 sr (Müller at al., 2007) would lead to better comparison with Raman retrievals. The second advantage of ElEx methodology is the ability to check the purity of the non-dust component when Raman retrievals are also available.

A 30 minute average backscatter coefficient analysis has been performed for the whole case day based on Klett-Fernald algorithm, along with the respective particle depolarization retrievals. For validation purposes, we use the aerosol optical depth of level 2, version 3 AERONET sunphotometer data. The total columnar aerosol optical depth, is compared with the lidar-derived aerosol optical depth, retrieved from integrating the aerosol extinction profiles from 0.5 to 5.5 km. To retrieve the Cimel sunphotometer aerosol optical depth at 532 nm, we use the measured aerosol optical depth at 500 nm and the Ångström exponent between 440 and 870 nm. The results are presented in Figure 7. The results show reasonable agreement between 4 and 10 UTC. There is a noticeable difference between the two datasets after 11 UTC. This is probably due to larger aerosol load in the first 0.5 km, which is not considered in the lidar aerosol optical depth retrieval. This can be seen from the time evolution of the extinction coefficient at the same day presented in Figure 7 (b). The plot shows that after 11 UTC there is enhanced aerosol load between 0.5 and 1 km. Thus, there is indication that the aerosol load could be also enhanced in the lower part of the atmosphere between 0 and 0.5 km. Although the fact that in these height ranges AOD cannot be retrieved due to incomplete overlap, the observed small differences could be attributed to the large aerosol load in the lower boundary

layer. The possibility of the retrieval of extinction profiles during day-time with a one-wavelength lidar system is the third advantage of ElEx methodology.

The quality of the extinction coefficient retrieved by ElEx is depending on the validity of the assumptions made. Firstly, an appropriate lidar ratio assumption is crucial for the retrieval of a realistic backscatter coefficient. In the case under study the assumption of 40 sr is reasonable taking into account the mixing state of the atmosphere. This is also confirmed by the Raman backscatter coefficient retrieval shown in Figure 8 (b). However, a different assumption of 20 sr (or 60 sr) (which are both extreme values for our case study) will increase (or decrease) the backscatter coefficient at height ranges between 1 and 2 km (Figure 8(a)). The difference would be smaller at higher levels. This assumption will afterwards affect the calculation of the particle depolarization (Figure 6(b)) and thus the contribution of the aerosol types involved (Figure 5). At ranges between 1 and 2 km a lidar ratio of 20 sr will modify the particle depolarization ratio from 25 to 20 %, while a selection of 60 sr will increase the particle depolarization ratio from 25 to 30 %. Thus, the choice of lidar ratio will affect the fraction of marine and dust particles calculated as shown in Figure 5. Changing the lidar ratio from 20 to 60 sr will result on differences of 40% for the contribution for dust particles in the lower part of the atmosphere. The proper selection of depolarization and lidar ratio values for vertical separation have been discussed in previous publications (e.g. Giannakaki et al., 2011, Gross et al., 2011). However, it should be strongly emphasized the critical choice of lidar ratio into the final fraction of the aerosol mixture when Tesche et al., 2008 method is used, especially in cases of elastic lidars.

In our particular case, the assumptions that were made proved to be reasonable, taken into account the trajectory analysis and the values usually observed for marine and dust particles, and thus the comparison with Raman retrievals shows only very small differences (Figure 2(c)). The application of the methodology in more complicated environments is also possible, giving however great attention on the aerosol mixing state and the initial values of the input parameters involved.

## 4 Conclusions

In this study we present a new method (ElEx) for the estimation of extinction coefficient lidar profiles with elastic depolarization lidars. ElEx make uses of the elastic backscatter coefficient in combination with depolarization ratio in the same wavelength, along with values of depolarization ratio and lidar ratio based on the literature. Reasonable agreement was found, both with Raman retrievals during night time, and with Cimel sunphotometer observations in terms of aerosol optical depth, during day time. This method can be only applied in stations with well-defined aerosol mixtures. There are several advantages of the proposed methodology. ElEx is not limited to nighttime Raman observations, and thus is applicable to daytime lidar measurement, with small time period analysis. In case of Raman extinction profiles availability the purity of non-dust component can be additionally checked. In stations with more complicated aerosol mixtures within the planetary boundary layer the methodology is possible to be applied to free tropospheric aerosol layers. CALIPSO extinction retrieval could be also improved when the aerosol mixture is known. However, attention should be given when the separation method is used in elastic backscatter retrievals, regarding not only the depolarization ratio values but also the lidar ratio values selected.

## Data Availability

Lidar data is available upon request from the authors and data quicklooks are available on PollyNET website (http://polly.tropos.de/). Trajectories are calculated with the NOAA (National Oceanic and Atmospheric Administration) HYSPLIT (HYbrid Single-Particle Lagrangian Integrated Trajectory) model (https://ready.arl.noaa.gov/HYSPLIT.php, accessed: 30/04/2019). BSC-DREAM8b model simulations are operated by the Barcelona Supercomputing Center and are available at https://ess.bsc.es/bsc-dust-daily-forecast/ (accessed: 30/04/2019). RAMS model can be found at ftp://ftp.mg.uoa.gr/pub/data/iclams1.3.tbz.

## Author contribution

EG developed the methodology and wrote the manuscript. EG, PK and EM performed the lidar data analysis. NB analyzed the model simulations. VA and AA initiated the measurement campaign. All authors participated in scientific discussions on this study and reviewed/edited the manuscript during its preparation process.

## Competing interests

The authors declare that they have no conflict of interest.

## Acknowledgments

The CHARADMexp campaign was funded from the ESA-ESTEC project "Characterization of Aerosol mixtures of Dust And Marine origin" contract no. IPL-PSO/FF/lf/14.489. The support of the Academy of Finland (project no. 270108) is greatly acknowledged. BSC-DREAM8b simulations were performed on the Mare Nostrum supercomputer hosted by the Barcelona Supercomputing Center-Centro Nacional de Supercomputación (BSC). Eleni Marinou acknowledges the support of the Deutscher Akademischer Austauschdienst (grant no. 57370121). Vassilis Amiridis acknowledges the support of the European Research Council (ERC) under the European Community's Horizon 2020 research and innovation framework program – ERC grant agreement 725698 (D-TECT). We acknowledge the support of "PANhellenic infrastructure for Atmospheric Composition and climatE chAnge" (MIS 5021516), which is implemented under the action "Reinforcement of the Research and Innovation Infrastructure", funded by the operational program "Competitiveness, Entrepreneurship and Innovation" (NSRF 2014–2020) and co-financed by Greece and the European Union (European Regional Development Fund). NOA team acknowledges the support of Stavros Niarchos Foundation (SNF).

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

**Figures**

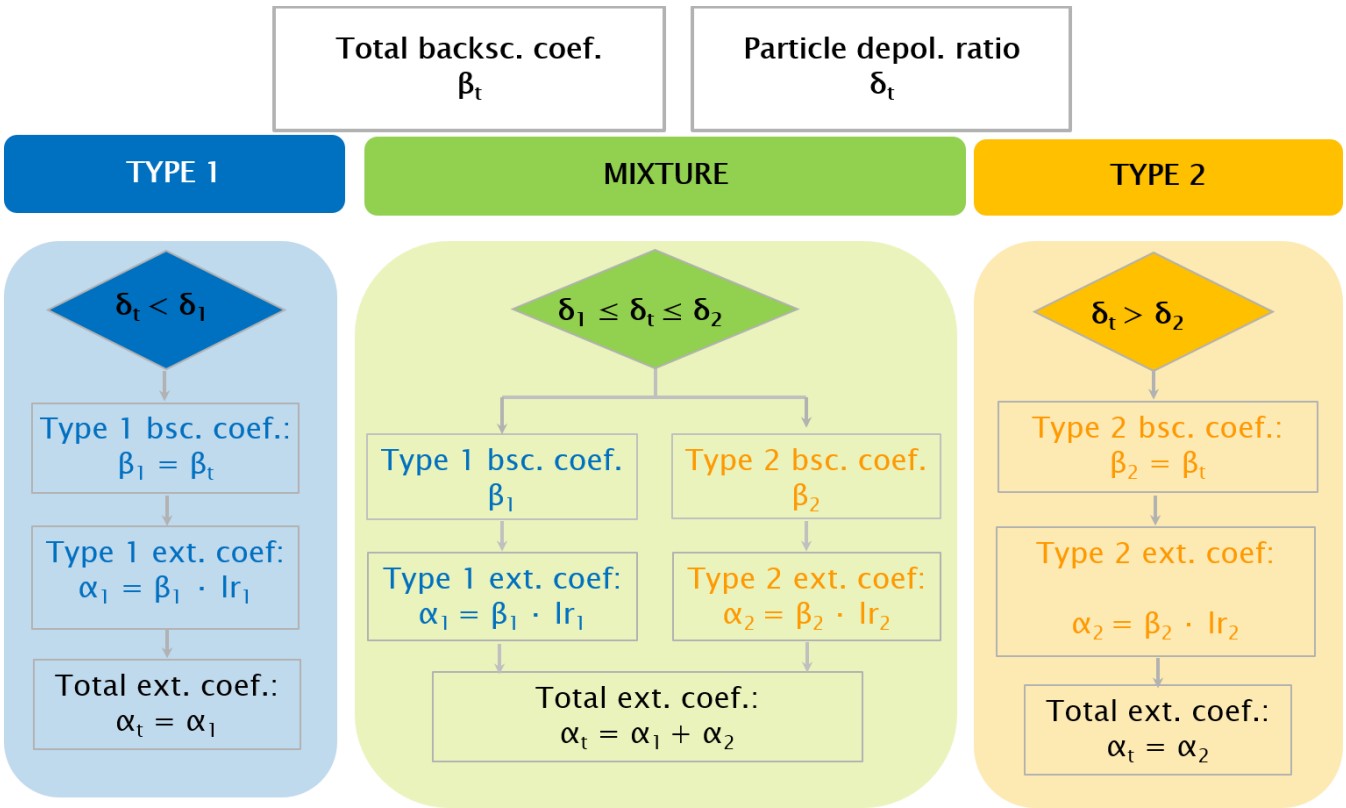

**Figure 1 Flowchart of the proposed methodology for the retrieval of extinction coefficient profile using backscatter and depolarization data.**



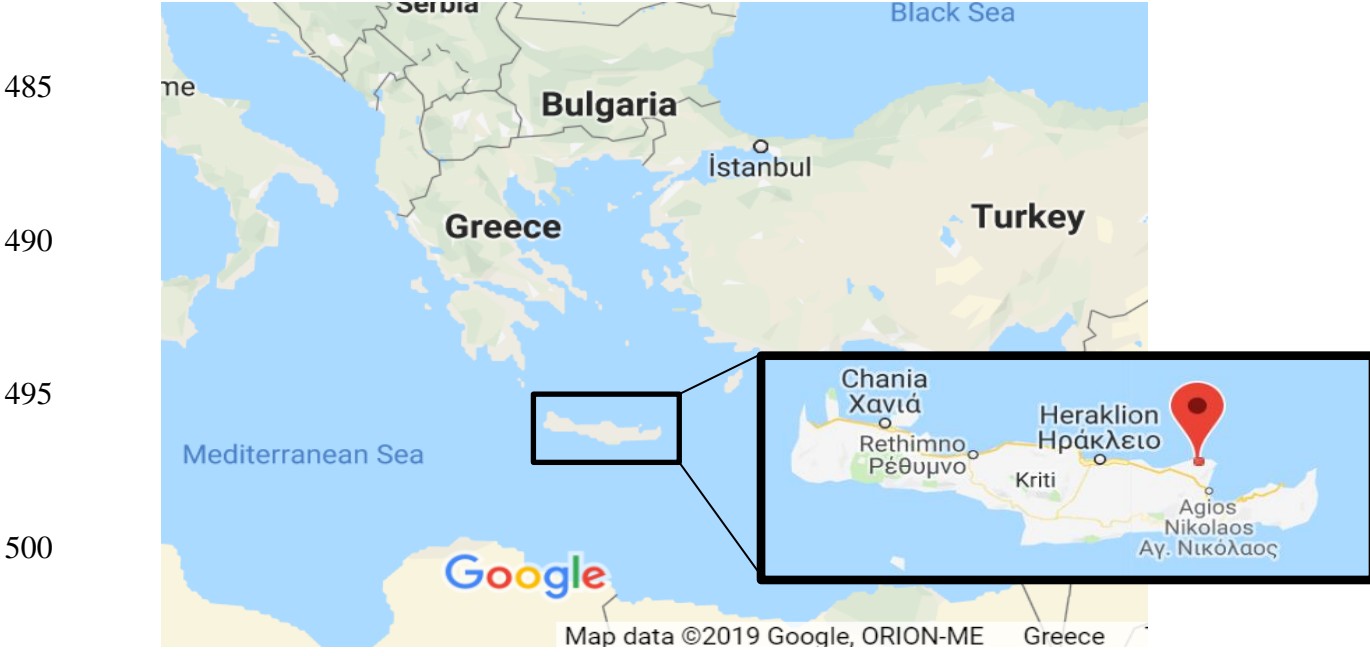





Google

Map data ©2019 Google, ORION-ME    Greece


**Figure 2. Location of Finokalia Station (red dot) in Crete, Greece.**

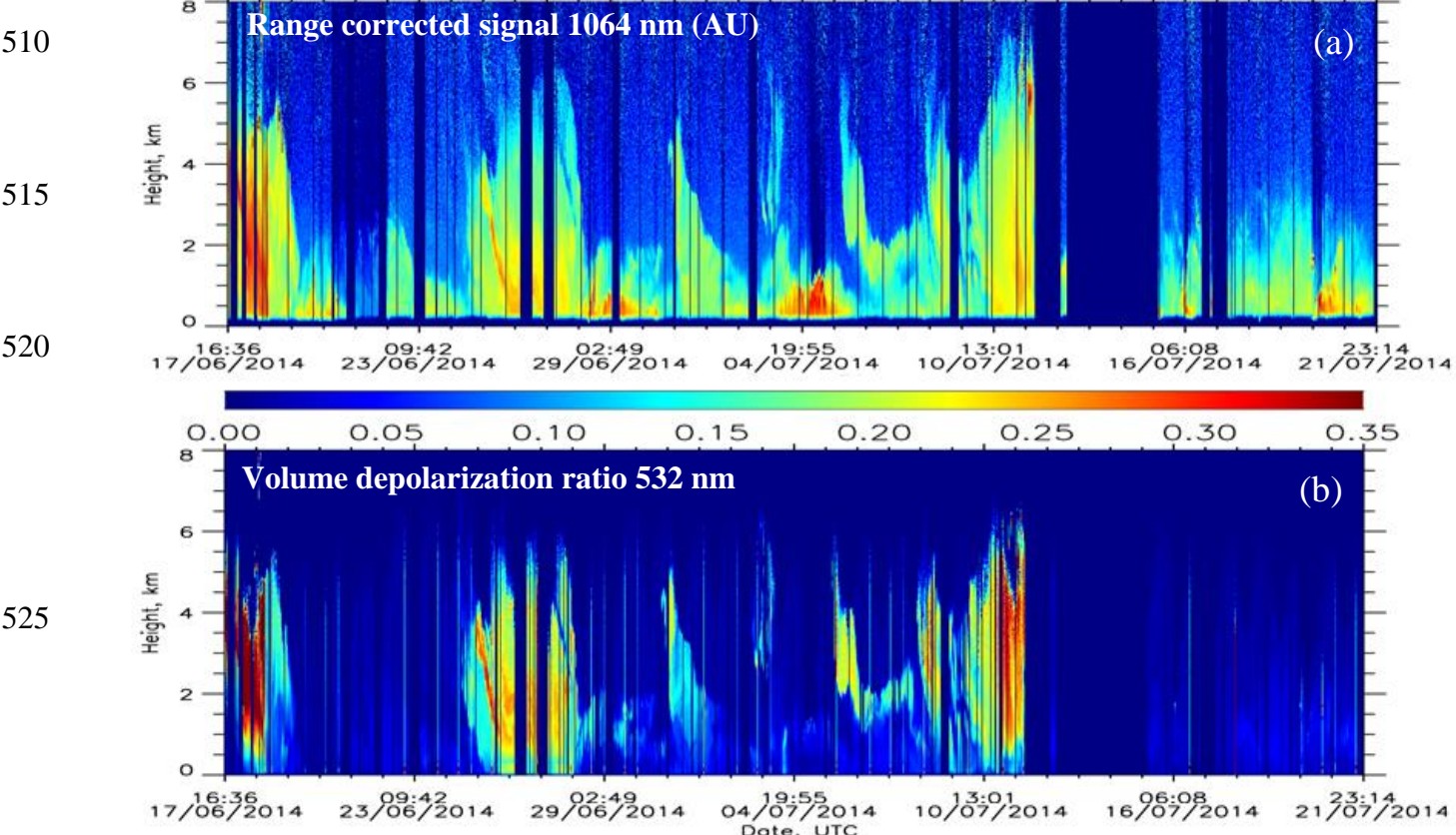





**Figure 3 Range corrected signal at 1064 nm (a) and volume depolarization ratio (b) at Finokalia during CHARADMExp campaign.**


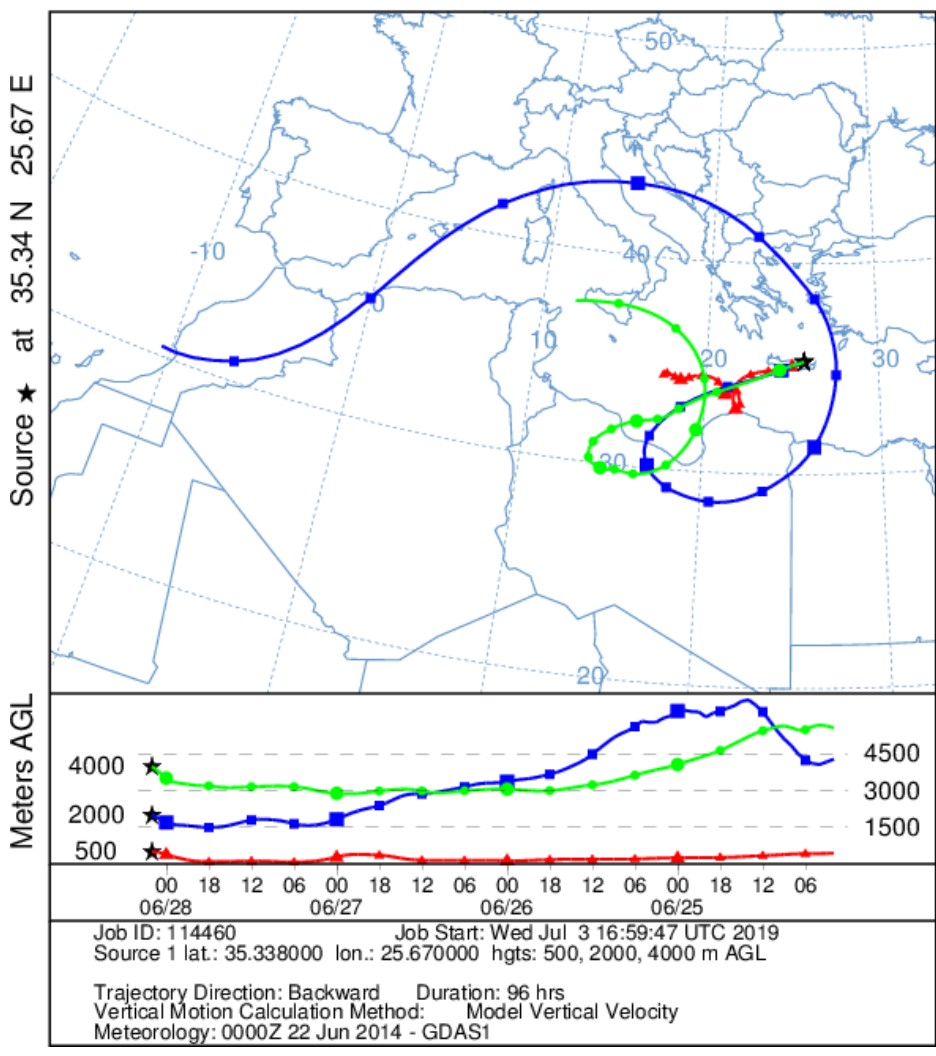

**Figure 4. Four days backward trajectories at 02:00 UTC on 28th June 2014 at Finokalia, Crete. The arrival heights are set to 500, 1000 and 4000 m.**

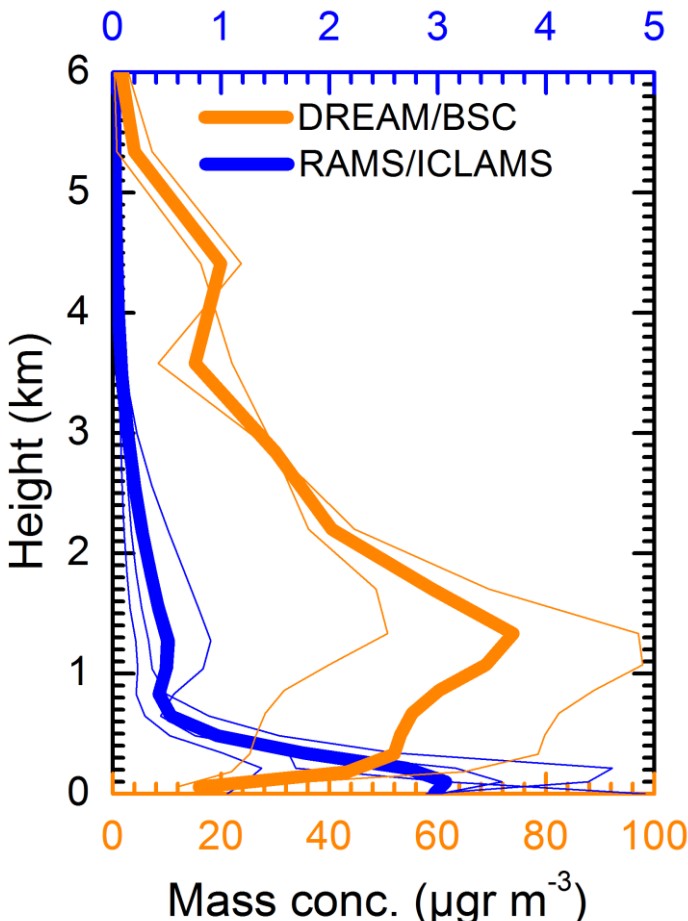

Figure 5. Simulations of sea-salt (blue) and dust (orange) mass concentrations profiles using RAMS/ICLAMS and DREAM/BSC models for 28th of June 2014, 00:00-03:00 UTC.

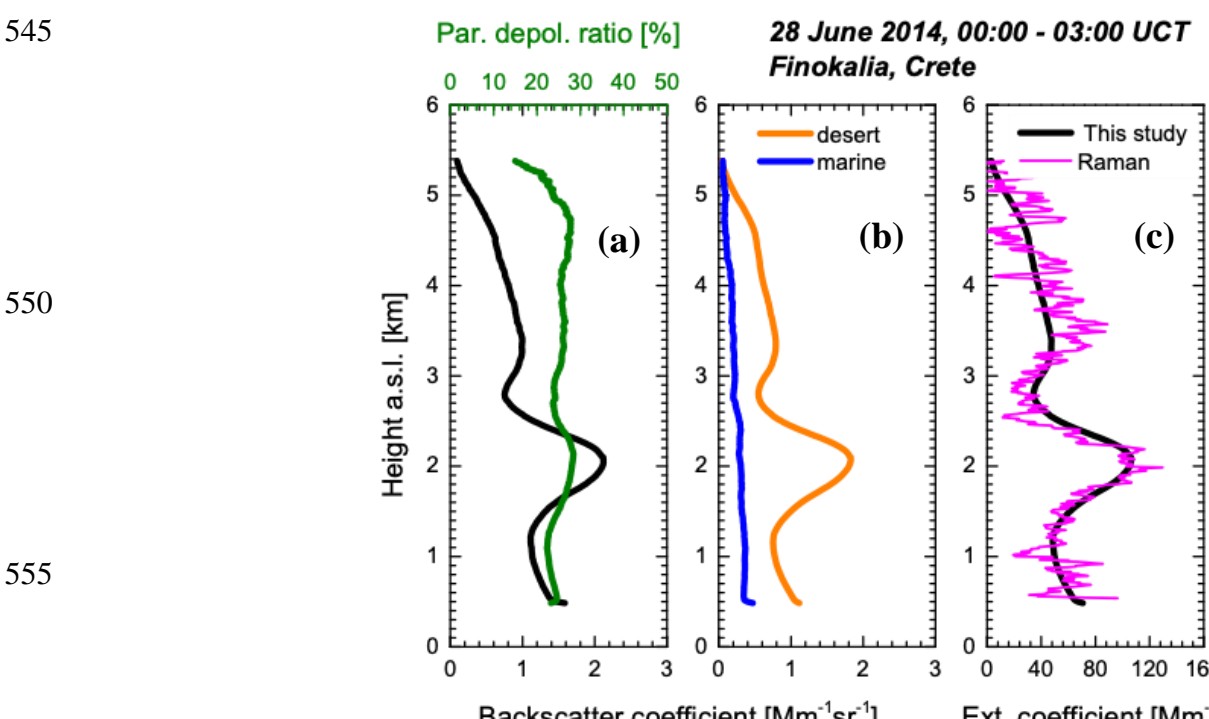

Figure 6: (a) Backscatter coefficient (black) and particle depolarization ratio (green) at 532nm for 28th of June 2014, 00:00-03:00 UTC. (b) Backscatter coefficient for desert dust (orange) and marine (blue) particles. (c) Extinction coefficient at 532 nm with the proposed method (black) and using the standard Raman retrieval (pink).

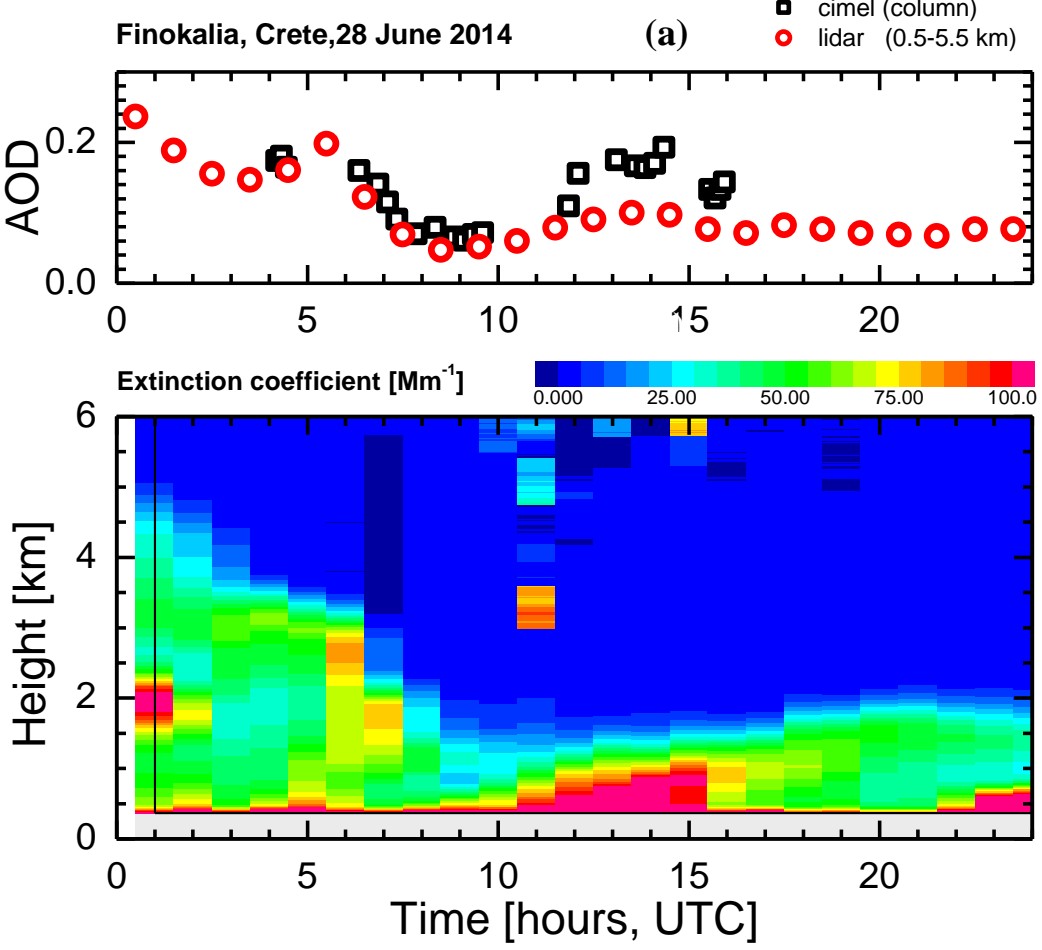

Figure 7. Time series of the aerosol optical depth at 532 nm from sun photometric data (total column) and from lidar retrieval (0.5-5.5 km). Color plot of the extinction coefficient for 28th of June 2014.

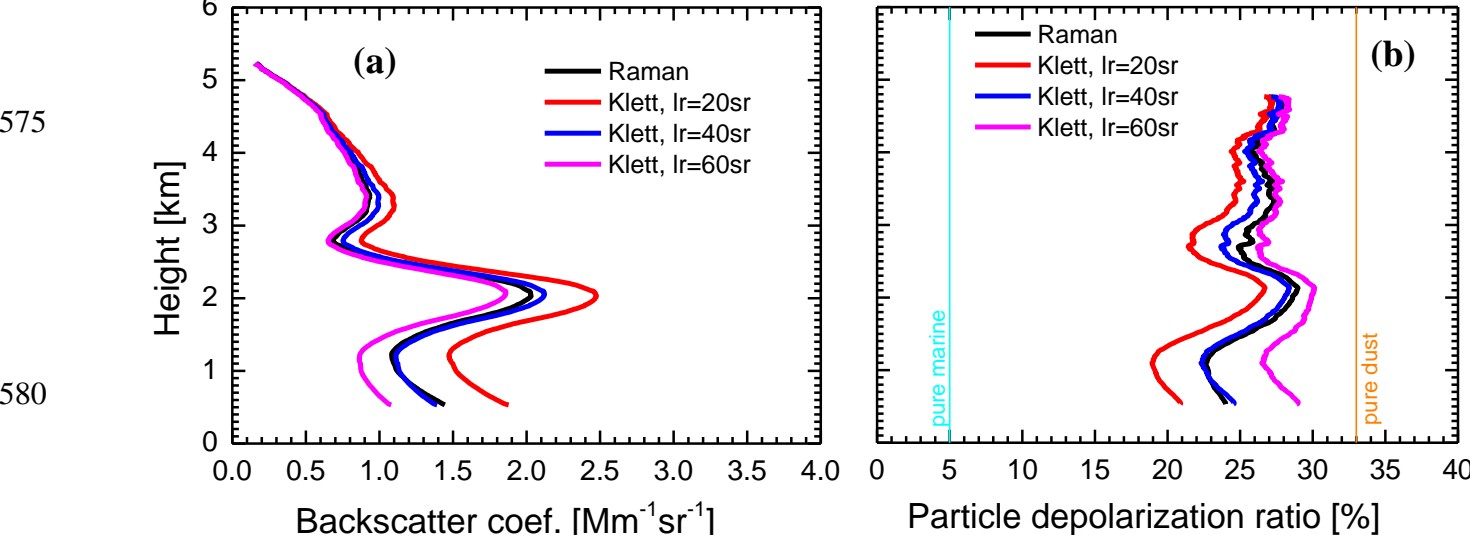

**Figure 8. (a)The backscatter coefficient for different lidar ratio initial values for Klett and Raman retrieval and (b) the corresponding particle depolarization ratio profiles.**

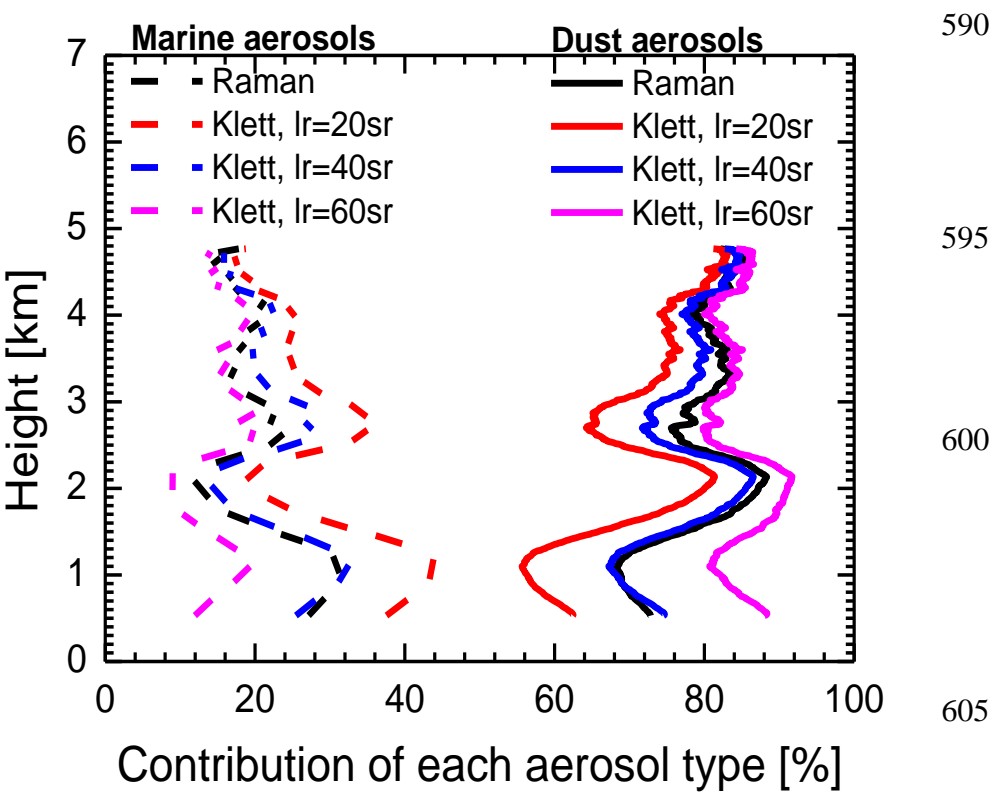

**Figure 9. Fraction of marine (dotted) and dust (solid) aerosols based on lidar ratio selection on Klett retrieval and the fraction calculated based on the Raman based technique (black).**

**Tables**

**Table 1. Overview of the input parameters used in our study.**

| Parameter | Aerosol Type | Symbol | Value | Reference |
|---|---|---|---|---|
| Lidar ratio at 532 nm | -Marine | $lr_m$ | 25 sr | Müller et al., 2007; Dawson et al. 2015; Gross et al. 2011; Gross et al. 2013, Gross et al. 2015 |
| | Saharan dust | $lr_d$ | 55 sr | Mattis et al. 2002; Mona et al. 2006; Papayannis et al. 2005; Balis et al. 2004, Gross et al. 2011; Gross et al. 2013, Gross et al. 2015 |
| Particle depolarization   ratio at 532 nm | Marine | $\delta_m$ | 0.05 | Gross et al. 2011; Gross et al. 2013, Gross et al. 2015 |
| | Saharan dust | $\delta_d$ | 0.31 | Freudenthaler et al. (2009); Gross et al. 2011; Gross et al. 2013, Gross et al. 2015 |

 **Tables**

**Table 1. Overview of the input parameters used in our study.**

| Parameter | Aerosol Type | Symbol | Value | Reference |
|---|---|---|---|---|
| Lidar ratio at 532 nm | Marine | $lr_m$ | 25 sr | Müller et al., 2007; Dawson et al. 2015; Gross et al. 2011; Gross et al. 2013, Gross et al. 2015 |
| | Saharan dust | $lr_d$ | 55 sr | Mattis et al. 2002; Mona et al. 2006; Papayannis et al. 2005; Balis et al. 2004, Gross et al. 2011; Gross et al. 2013, Gross et al. 2015 |
| Particle depolarization ratio at 532 nm | Marine | $\delta_m$ | 0.05 | Gross et al. 2011; Gross et al. 2013, Gross et al. 2015 |
| | Saharan dust | $\delta_d$ | 0.31 | Freudenthaler et al. (2009); Gross et al. 2011; Gross et al. 2013, Gross et al. 2015 |