# Peer review of "The potential of elastic/polarization lidars to retrieve extinction profiles"

_Atmospheric Measurement Techniques, 2019_

## Referee Comment (RC1) · Anonymous Referee #2 · 14 Nov 2019

The authors show a study for estimating particle extinction profiles based on elastic and polarization channels from a lidar system. The data used were acquired during a field campaign in Crete. The methodology employed is well known and a direct comparison among Raman profiles and the particle linear depolarization is perfomed. While the results are quite insteresting I consider it would be necessary to stress the novelty contribution given by this paper. Apparently it is the ElEx approach but in that case I would give more detail to the description of the methodology and discuss better if using a Raman signal is a good validation method specially when concerning uncertainties of this approach. Apparently one of the conclusions is a strong dependecy on the lidar ratio value and in this context it is important to check the uncertainties involved in the process.

[Figure]

Section 2.4 Line 115 – Extinction Section 2.4 Line 118 – do not have Section 2.4 Line 118 – Only at a later stage, Conclusions Line 227 - EIEx

Please improve Figure 7 , Figure 8 and Figure 9 resolutions

---

## Referee Comment (RC2) · Anonymous Referee #3 · 19 Nov 2019

The paper presents a methodology for the estimation of aerosol extinction profiles form single wavelength elastic lidars with a depolarization channel. It is presented the general concept and its applicability is examined. The paper is appropriate for AMT and the methodology is very useful for daytime measurements of simple lidar systems. The paper should be accepted for publication after considering my comments below.

Abstract. To my understanding the focus of the paper is to present the Elastic Extinction Retrieval (EER) methodology, using data at Finokalia as a case study. Therefore the abstract should be rephrased accordingly.

Line 25. Please quantify what is reasonable difference.

Line 33. Please replace "The types of aerosols" with "Pure aerosol types"

[Figure]

Line 35 Replace "is occurred" with "occur" and in the references use e.g.

Line 37. Add "the" before the European.

Line 38. Please provide a reference for your claim that there is no mixing between maritime and desert dust.

Lines 52 to 64. The authors should mention here that many of the references correspond to the POLIPHON algorithm. In addition they should outline here that in this paper they aim to present a general concept algorithm approach rather than presenting cases studies as they did in their 2017 papers.

Section 2. To my understanding the main focus of this section should be 2.4 which should be rephrased as "Description of the EER method. This section should be written in a general way and not dependent on the certain lidar system. I would suggest to expand this section, and move earlier as 2.1. In addition they authors should provide the equations they use so that the reader can reproduce their methodology. In addition I would suggest that they should not limit the description of the methodology to mixtures of marine and dust but in general for mixtures of aerosols with distinct intensive parameters. They should also examine how sensitive is their algorithm to the a priori lidar ratio values considered. Of course then they can continue with the Finokalia case study as demonstration case study.

---

## Author Comment (AC1) · 23 Dec 2019

We would like to thank the two reviewers for devoting time to read our manuscript and provide valuable comments for improving it and increasing its scientific value. We have modified our manuscript following the guidelines given by the two reviewers.

Kind regards,

Giannakaki Elina, on behalf of all the co-authors

Interactive comment of RC1 on Atmos. Meas. Tech. Discuss., doi:10.5194/amt-2019-271, 2019.

Below we answer to 1$^{st}$ reviewer's comment (RC). The **RC**s are given in **bold**, our replies in plain font and the corresponding *changes in the manuscript* are given in *italic*.

**The authors show a study for estimating particle extinction profiles based on elastic and polarization channels from a lidar system. The data used were acquired during a field campaign in Crete. The methodology employed is well known and a direct comparison among Raman profiles and the particle linear depolarization is perfomed. While the results are quite insteresting I consider it would be necessary to stress the novelty contribution given by this paper. Apparently it is the ElEx approach but in that case I would give more detail to the description of the methodology and discuss better if using a Raman signal is a good validation method specially when concerning uncertainties of this approach. Apparently one of the conclusions is a strong dependecy on the lidar ratio value and in this context it is important to check the uncertainties involved in the process.**

We would like to thank the Anonymous Referee 2 for the constructive comments and recommendations. We rearranged the methodology section in order to generalize the application of our methodology as suggested. In such a way more emphasis has been given to the novelty contribution of this study. The method is not based in a case study of desert dust and marine aerosol mixture but in Type 1 and Type 2 in general. The flowchart of ElEx is given in Figure 1 and a discussion of the pure aerosol types that can be observed along with their intensive optical properties is given in subsection 2.2.3. In addition in section 2.2.2 and 2.2.3 we provide the equations so that the reader can reproduce ElEx methodology. In section 2.2 we present our lidar system and data processing. Our case study of marine and dust is given in detail in subsection 3.2 and the sensitivity analysis of the lidar ratio is part of subsection 3.2.

**Section 2.4 Line 115 – Extinction Section**
"Elastic Extinxtion" has been replace by "Elastic Extinction".

*A new method, called ElEx [= Elastic Extinction], is proposed for the estimation of extinction coefficient lidar profiles using only the information provided by the elastic and polarization channels of a lidar system.*

**2.4 Line 118 – do not have**
The structure of the methodology section has been changed as proposed by the reviewer 3. The sentence "Initially, we assume that we do not have the PollyXT Raman channel" has been deleted.

**Section 2.4 Line 118 – Only at a later stage**

The structure of the methodology section has been changed as proposed by the reviewer 3. The sentence in now in section 2.2 written differently.

*Extinction coefficient profiles at 532 nm are also retrieved based on the Raman method (Ansmann et al., 1992) and are only used for validation purposes of the proposed methodology at a later stage in this study.*

**Conclusions Line 227 - ElEx**

"Elex" has now been corrected to "ElEx".

*ElEx is not limited to nighttime Raman observations, and thus is applicable to daytime lidar measurement, with small time period analysis.*

**Please improve Figure 7 , Figure 8 and Figure 9 resolutions**

Thank you the reviewer for this comment! We have improved the resolution of Figure 7,8 and 9.

---

## Author Comment (AC2) · 23 Dec 2019

We would like to thank the two reviewers for devoting time to read our manuscript and provide valuable comments for improving it and increasing its scientific value. We have modified our manuscript following the guidelines given by the two reviewers.

Kind regards,

Giannakaki Elina, on behalf of all the co-authors

Interactive comment of RC2 on Atmos. Meas. Tech. Discuss., doi:10.5194/amt-2019-271, 2019.

We would like to thank the Anonymous Referee 2 for the constructive comments and recommendations. We rephrased the abstract and the methodology section in order to generalize the application of our methodology as suggested. More emphasis has been also given to lidar ratio selection as suggested by the two reviewers. Below we answer to $2^{nd}$ reviewer's comment (RC). The **RC**s are given in **bold**, our replies in plain font and the corresponding *changes in the manuscript* are given in *italic*.

**Abstract. To my understanding the focus of the paper is to present the Elastic Extinction Retrieval (EER) methodology, using data at Finokalia as a case study. Therefore the abstract should be rephrased accordingly.**

Indeed, the focus of the paper is to present the Elastic Extinction Retrieval, providing the advantages and disadvantages, using a case study at Finokalia. So, we rephrased the abstract accordingly.

*A new method, called ElEx, is proposed for the estimation of extinction coefficient lidar profiles using only the information provided by the elastic and polarization channels of a lidar system. The method is applicable both during day-time and night-time lidar measurements under well-defined aerosol mixtures. When the two aerosol type have different optical properties, permit the separation of the aerosol mixture. ElEx uses the particle backscatter profiles at 532 nm and the vertically resolved particle linear depolarization ratio measurements at the same wavelength. The particle linear depolarization ratio and the lidar ratio values of pure aerosol types are taken from literature. The total extinction profile is then estimated and compared well with Raman retrievals. In this study, ElEx was applied in an aerosol mixture of marine and dust particles at Finokalia station during CHARADMEx campaign. Any difference between ElEx and Raman extinction profiles indicates that the non-dust component could be probably attributed to polluted marine or polluted continental aerosols. Comparison with sun-photometric aerosol optical depth observations is performed as well during daytime. Differences in the total aerosol optical depth is varying between 1.2 and 72% and is attributed to the limited ability of the lidar to correctly represent the aerosol optical properties in the near range due to overlap problem.*

**Line 25. Please quantify what is reasonable difference.**

We deleted the word reasonable, and instead we give information about the calculated difference.

*Differences in the total aerosol optical depth is varying between 1.2 and 72% and is attributed to the limited ability of the lidar to correctly represent the aerosol optical properties in the near range due to overlap problem.*

**Line 33. Please replace "The types of aerosols" with "Pure aerosol types"**
The phrase "The types of aerosols" has been replace by "Pure aerosol types".

*Pure types of aerosols can be categorized roughly as mineral dust, sea salt, volcanic, carbonaceous, or sulfate aerosols originating from various natural and anthropogenic sources.*

**Line 35 Replace "is occurred" with "occur" and in the references use e.g.**
The phrase "is occurred" has been replaced by "occur". We used e.g. in the references provided.

*Several lidar studies have revealed that a broad variety of aerosol mixtures occur in the European continent (e.g. Balis et al., 2004, Papayannis et al. 2005)*

**Line 37. Add "the" before the European.**
We added "the" before the European.

*The mixing occurs because of the relatively long pathways of air masses across different aerosol source regions before the detection over the European continent.*

**Line 38. Please provide a reference for your claim that there is no mixing between maritime and desert dust.**
The phrase is confusing. We deleted the phrase "For example, Saharan dust observed in South Europe is often already lifted over Africa to heights above 1-2 km, so that mixing with marine particles is almost prohibited".

**Lines 52 to 64. The authors should mention here that many of the references correspond to the POLIPHON algorithm. In addition they should outline here that in this paper they aim to present a general concept algorithm approach rather than presenting cases studies as they did in their 2017 papers.**
The POLIPHON is used for separation of dust and non dust backscatter contributions and the new, extended approach to separate even the fine and coarse dust backscatter fractions, while ElEx is a method to calculate the extinction profile with an Elastic lidar. However, both of them are using the separation technique and so we added a sentence about the POLIPHON algorithm. Also, more emphasis is given about the contribution of this study.

*The technique is the base of POLIPHON algorithm for ground –based lidars (Ansmann et al., 2019) and has also been applied to CALIPSO aerosol profiles either on selected case studies (Giannakaki et al., 2011) or on a statistical basis (Marinou et al., 2017).*
*In this contribution, we propose a method to determine the extinction coefficient profile using only the elastic and polarization lidar channels at 532 nm in a well-defined aerosol mixture. The method has been first suggested by Giannakaki et al. (2017) and further applied by Ansmann et al. (2017). In this contribution, we fully outline the methodology providing an extended sensitivity analysis along with the main advantages and limitations of it.*

**Section 2. To my understanding the main focus of this section should be 2.4 which should be rephrased as "Description of the EER method. This section should be written in a general way and not dependent on the certain lidar system. I would suggest to expand this section, and move earlier as 2.1. In addition they authors should provide the equations they use so that the reader can reproduce their methodology. In**

addition I would suggest that they should not limit the description of the methodology to mixtures of marine and dust but in general for mixtures of aerosols with distinct intensive parameters. They should also examine how sensitive is their algorithm to the a priori lidar ratio values considered. Of course then they can continue with the Finokalia case study as demonstration case study.

Thank you the reviewer for this comment. We rearranged the methodology section. It is now 2 subsections. In section 2.1 we present the general methodology that is not based either on the lidar system nor on the specific case study. The methodology is given 3 steps (2.1 Backscatter and Depolarization retrievals, 2.2 Separation of backscatters and 2.3 Estimation of Extinction). The method is not presented based on a mixture of desert dust and marine aerosol mixture but based on Type 1 and Type 2 in general. The flowchart of ElEx is given in Figure 1 and a discussion of the pure aerosol types that can be observed along with their intensive optical properties is given in subsection 2.2.3. In addition we provide the equations so that the reader can reproduce ElEx methodology. In section 2.2 we present our lidar system and data processing. Our case study is given in detail in subsection 3.2 and the sensitivity analysis of the lidar ratio is part of subsection 3.2.

[revised manuscript text omitted]